# Exploring the Mediators that Promote Carotid Body Dysfunction in Type 2 Diabetes and Obesity Related Syndromes

**DOI:** 10.3390/ijms21155545

**Published:** 2020-08-03

**Authors:** Joana F. Sacramento, Kryspin Andrzejewski, Bernardete F. Melo, Maria J. Ribeiro, Ana Obeso, Silvia V. Conde

**Affiliations:** 1CEDOC (Chronic Disease Research Center), NOVA Medical School, Faculdade de Ciências Médicas, Universidade Nova de Lisboa, 1150-082 Lisbon, Portugal; joana.sacramento@nms.unl.pt (J.F.S.); kandrzejewski@imdik.pan.pl (K.A.); bernardete.melo@nms.unl.pt (B.F.M.); mj.rfribeiro@gmail.com (M.J.R.); 2Department of Respiration Physiology, Mossakowski Medical Research Centre, Polish Academy of Sciences, Pawińskiego 5, 02-106 Warsaw, Poland; 3Departamento de Bioquímica y Biología Molecular y Fisiología, Universidad de Valladolid, Facultad de Medicina, Instituto de Biología y Genética Molecular, CSIC, Ciber de Enfermedades Respiratorias, CIBERES, Instituto de Salud Carlos III, 47005 Valladolid, Spain; aobeso@ibgm.uva.es

**Keywords:** carotid body, obesity related syndromes, type 2 diabetes, glucose, insulin, leptin, inflammation, sympathetic overactivation

## Abstract

Carotid bodies (CBs) are peripheral chemoreceptors that sense changes in blood O_2_, CO_2_, and pH levels. Apart from ventilatory control, these organs are deeply involved in the homeostatic regulation of carbohydrates and lipid metabolism and inflammation. It has been described that CB dysfunction is involved in the genesis of metabolic diseases and that CB overactivation is present in animal models of metabolic disease and in prediabetes patients. Additionally, resection of the CB-sensitive nerve, the carotid sinus nerve (CSN), or CB ablation in animals prevents and reverses diet-induced insulin resistance and glucose intolerance as well as sympathoadrenal overactivity, meaning that the beneficial effects of decreasing CB activity on glucose homeostasis are modulated by target-related efferent sympathetic nerves, through a reflex initiated in the CBs. In agreement with our pre-clinical data, hyperbaric oxygen therapy, which reduces CB activity, improves glucose homeostasis in type 2 diabetes patients. Insulin, leptin, and pro-inflammatory cytokines activate the CB. In this manuscript, we review in a concise manner the putative pathways linking CB chemoreceptor deregulation with the pathogenesis of metabolic diseases and discuss and present new data that highlight the roles of hyperinsulinemia, hyperleptinemia, and chronic inflammation as major factors contributing to CB dysfunction in metabolic disorders.

## 1. Introduction

Metabolic diseases such as obesity, metabolic syndrome, and type 2 diabetes are some of the most common non-communicable diseases whose prevalence continues to increase, contributing to significant morbidity and mortality worldwide and considered worldwide epidemics [1,2]. The increasing incidence of these diseases is mainly due to lifestyle changes such as the sedentary lifestyle and the increase in the consumption of hypercaloric diets. The sympathetic nervous system is known to play a role in the generation of metabolic diseases [3,4] and several factors have been postulated to be responsible for this increased sympathetic activation, such as hyperinsulinemia, hyperleptinemia, and inflammatory cytokines [5]. However, there is no consensus on the major mediators responsible for it. Additionally, afferent pathways and the stimuli that trigger afferent activation are poorly studied.

Carotid bodies (CBs), located bilaterally at the bifurcation of each common carotid artery, are peripheral chemoreceptors that classically sense changes in arterial blood O_2_, CO_2_, and pH levels. In response to hypoxia (O_2_ deprivation), hypercapnia (CO_2_ retention), and acidosis (pH drop), type I cells, the CB chemosensory unit, release neurotransmitters that act on the nerve terminals of the CB sensitive nerve, the carotid sinus nerve (CSN), to generate action potentials or to inhibit its activity [6]. CSN activity is integrated in the brainstem to induce a set of respiratory reflexes aimed, primarily, at normalizing the altered blood gases via hyperventilation [6] and regulating blood pressure and cardiac performance via sympathetic nervous system activation [7]. Besides its role as an oxygen sensor, in the last few years, the CB has also been proposed to be a metabolic sensor implicated in the control of carbohydrate and lipid metabolism [8,9,10,11] and in the regulation of peripheral insulin sensitivity and glucose homeostasis [8,9,12,13,14,15,16,17] (Figure 1a). Recently, we showed that CB activity is increased in prediabetes and type 2 diabetes animal models [13,18,19,20] and patients [21] (Figure 1a) and that the abolishment of CB activity in animals, via chronic resection of the CSN or CB ablation, prevents and reverses dysmetabolism in rodent models of metabolic disease [13,15,16] (Figure 1b) by positively impacting glucose uptake and insulin signaling in the liver and in the visceral adipose tissue [15]. Additionally, we have previously shown that CSN resection in animal models prevents and restores the heightened sympathetic activity, measured as increased plasma and adrenal medulla catecholamines levels and increased LF bands and LF/HF ratio in heart rate variability analysis, which is characteristic of metabolic diseases [13,15]. In agreement with this heightened sympathetic activity, electrophysiological recordings at the superior cervical chain confirmed the overactivation of the sympathetic nervous system in rodent models of dysmetabolism that disappear with CSN resection [19]. Altogether, these results mean that CB dysfunction is involved in the development of metabolic diseases via an overactivation of the sympathetic nervous system (Figure 1a).

Alongside the hypothesis that CB dysfunction contributes to metabolic diseases, we described how hypercaloric diet animals [13] and prediabetes patients [21] exhibit increased basal ventilation and showed that prediabetic dysmetabolism correlates with increased peripheral chemosensitivity, as evaluated by the Dejours test, which measures the decrease in basal ventilation produced by 100% O_2_ (hyperoxia), and that this correlates with abdominal perimeter and insulin resistance [21]. This increased peripheral chemosensitivity observed both in prediabetes and type 2 diabetes was shown to be associated with increased CB weight, with an increase in the percentage of CB chemoreceptor cells and CB tyrosine hydroxylase activity as well as with augmented CB release of catecholamines and CSN activity in metabolic disease animals [13,18,19,22] and with an increase in CB size of approximately 25% in type 2 diabetes patients [23]. In line with our hypothesis that CB dysfunction is associated dysmetabolism, it was shown by Paleczny et al. that, in men, overweight/obesity is accompanied by an augmented blood pressure response from peripheral chemoreceptors, while respiratory and heart rate responses remain unaltered, and that hyperinsulinemia and insulin resistance (but not hyperleptinemia) are associated with an augmented pressure response from chemoreceptors [24]. In agreement with the pre-clinical data showing that CB dysfunction is involved in the development of metabolic diseases, hyperbaric oxygen therapy, an intervention that dramatically reduces CB activity [25], improves fasting glucose and post-prandial glucose management in type 2 diabetes patients [26] (Figure 1b).

Besides prediabetes and type 2 diabetes, CB overactivation has been associated with other pathologies that occur with cardiometabolic comorbidities, such as obstructive sleep apnea (OSA) [27,28]. OSA is a potentially serious sleep disorder characterized by repetitive episodes of complete or partial nocturnal breathing obstructions and by altered hypoxic ventilatory responses [27,29,30,31], being also associated with several metabolic and cardiovascular abnormalities [32,33,34]. The consensus is that CB overactivation due to chronic intermittent hypoxia (CIH) is involved in the genesis of OSA-mediated hypertension and insulin resistance through an increase in sympathetic nervous system activity [10,27,29,30,35]. One of the risk factors contributing to the development of OSA is obesity and, in fact, it is estimated that 40% of obese individuals have OSA and that approximately 70% of individuals with OSA are obese [36,37]. In addition, in middle-aged men with OSA, it was observed that there is an association with visceral obesity, inflammation, hyperinsulinemia, and hyperleptinemia [36]. Obesity is also known to be present in 90% of type 2 diabetes patients and in 85–99% of metabolic syndrome patients, being a risk factor for the development of these diseases [38,39] due to adipose tissue dysfunction. White adipose tissue is the main lipid storage depot in humans, being critically important in buffering the dietary fat influx entering the circulation by suppressing the release of non-esterified fatty acids into the circulation and by increasing the clearance of triacylglycerols [40]. Apart from the decrease in lipid storage and the release of free fatty acids into the circulation, promoting thereby whole-body insulin-resistance, adipose tissue plays a crucial role in the development of metabolic diseases, due to its critical immune and endocrine functions. Adipose tissue is an active endocrine organ that secretes several humoral factors named adipokines, such as leptin and adiponectin, that exhibit important systemic metabolic effects, from food intake to glucose tolerance [41,42,43]. Along with the production of specialized adipokines, such as leptin and adiponectine, adipose tissue also secretes proinflammatory cytokines that contribute to the low-level systemic inflammation that is involved in adipose tissue dysfunction and systemic metabolic abnormalities [43,44].

Leptin, also known as the satiety hormone, is increased in the plasma of obese subjects, being highly associated with the degree of adiposity and body mass index [45]. Apart from regulating satiety, leptin has other body functions and has been proposed to be one of the major contributors to the increased sympathetic nervous system activity observed in obesity and obesity-induced cardiometabolic disturbances. It was observed that there is a positive correlation between whole-body norepinephrine spillover and plasma leptin levels in overweight and obese metabolic syndrome subjects [46], results that were also described in animal studies [47]. This effect of leptin in promoting sympathetic nervous system activation has been mainly attributed to its action on the central nervous system [48]; however, we can postulate that actions outside the brain can add to these effects. Like leptin, proinflammatory cytokines, such as tumor necrosis factor alpha (TNF-α) and interleukin 6 (IL-6), have also been suggested to increase sympathetic nervous system activity by acting centrally [5]. Serum levels of TNF-α and IL-6 are increased in diabetic patients, being found in higher levels in obese than non-obese diabetic patients [49]. Moreover, central IL-6 administration promoted an increase in uncoupling protein 1 (UCP1) activity and oxygen consumption [50,51], suggesting a direct link to sympathetic nervous system activation. However, it cannot be ruled out that, besides its action in the central nervous system, IL-6 as well as other pro-inflammatory cytokines may exert their effects in periphery to increase sympathetic nervous system activity.

Another factor that has been shown to enhance sympathetic drive in cardiometabolic diseases is hyperinsulinemia [52], particularly by acting in the central nervous system. The injection of insulin into the arcuate nucleus and paraventricular nucleus promote an increase in spinal sympathetic outflow, mediated by the dorsal hypothalamus and rostral ventrolateral medulla (for a review, see [53]). Additionally, insulin infusion augments muscle sympathetic nervous activity in healthy individuals under euglycemic conditions [54,55]. Nevertheless, these effects cannot be exclusively attributed to central nervous system-mediated mechanisms, as the administration of insulin into the carotid artery of anesthetized dogs leads to a higher increase in blood pressure and sympathetic activity than systemic insulin administration, this effect being abolished by the ganglionic blockade [56]. These results clearly suggest a role for the peripheral nervous system in insulin-mediated sympathetic activity.

Therefore, knowing that leptin, proinflammatory cytokines, and insulin augment sympathetic nervous system activity, that the CB integrates its action centrally via the modulation of sympathetic activity, and that the CB possesses receptors for all these mediators, we hypothesize that hyperleptinemia, proinflammatory cytokines, and hyperinsulinemia are mediators that contribute to the CB dysfunction that contributes to the genesis of metabolic diseases (Figure 2).

## 2. Insulin: A Stimulus for Carotid Body Activation in Metabolic Diseases

In the last few decades, the CB has been proposed to have metabolic sensing properties by responding to alterations in blood glucose and insulin [10,17]. Considering the anatomic location of the CB and its high vascularization [6], it is possible to anticipate that the CB is able to monitor metabolic states in the blood, this information being integrated in the brain.

Several in vivo and in vitro studies demonstrated the involvement of the CB in whole-body glucoregulation [8,9,12,57,58,59] and, more recently, it was postulated that it may directly sense arterial glucose concentrations [60,61]. However, to date, no consensus has been reached regarding CB’s glucose sensing properties and the involvement of hyperglycemia in the mechanisms promoting CB overactivation in metabolic diseases does not seem plausible [14]. In 2002, Pardal and López-Barneo reported in CB slices that hypoglycemia augmented chemoreceptor cells’ sensitivity and enhanced their response to hypoxia [60]. Additionally, in co-cultures of petrosal ganglions with CB chemoreceptor cells, Zhang et al. [61] reported that hypoglycemia increased afferent action potential frequency. On the other hand, Conde et al. [61] reported, using freshly isolates intact rat CB preparations, that hypoglycemia did not modify catecholamine release from CB type I cells or CSN action potential frequency [62], results confirmed previously by Almaraz et al. [62] and Bin-Jaliah et al. [63] obtained in intact CB-CSN preparations. More recently, Shirahata et al. [64] described in mice that basal CSN frequency of discharge and the CSN response to hypoxia did not change with low glucose concentrations, supporting the idea that CBs do not sense low glucose levels directly. Knowing that metabolic diseases are characterized by high glucose concentrations in blood, our group tested, in the isolated rat CB-CSN ex vivo preparation, the effect of 25mM of glucose on CSN chemosensory activity and showed that hyperglycemia did not change either basal or CSN chemosensory activity in response to hypoxia [14]. This suggests that hyperglycemia acting directly on the CB cannot be one of the major factors contributing to CB dysfunction in metabolic diseases.

However, even if the CB is not able to sense glucose directly, there is undoubtedly an association between CB activity and the regulation of glucose homeostasis. CB stimulation promoted a reflex hyperglycemia [65] and increased hepatic glucose output in cats [12], effects that were abolished by CSN resection [12]. The role of CB in the regulation of glucose homeostasis was also demonstrated in vivo, by Koyama et al. [8], since dogs submitted to CSN denervation exhibit a reduction in arterial glucagon in basal conditions. Additionally, the same authors reported that CB-resected dogs, in response to a hypoglycemia induced by insulin, exhibit a decrease in glucagon and cortisol levels, together with a reduction in endogenous hepatic glucose production in response to hypoglycemia and an increase in insulin sensitivity, independent of blood glucose level [8]. In agreement with the results obtained by Koyama et al. [8] in dogs, when CB activity was blunted by hyperoxia in healthy volunteers, there was a decrease in the release of counter-regulatory hormones such as epinephrine, cortisol, glucagon, and growth hormone in response to hypoglycemia induced by hyperinsulinemia [58]. Nevertheless, the counterregulatory responses to hypoglycemia in CB-resected patients were relatively normal [59]. In addition, in another clinical study, both hypoglycemia and hyperglycemia induced an increase in ventilation and in the ventilatory response to hypoxia [66], suggesting that, apart from glucose, insulin could account for the effects described above.

Bin-Jaliah et al. [63] reported that insulin-induced hypoglycemia increased spontaneous ventilation in rats, an effect that was abolished in CSN-sectioned animals, but that hypoglycemia per se was unable to alter CSN frequency, suggesting that the effects of insulin-induced hypoglycemia were mediated only by insulin and not by hypoglycemia. The effect of insulin per se on ventilation and on CB activity was then confirmed in 2013 [13]. It was observed in anesthetized rats that insulin, during an euglycemic clamp, increased ventilation in a dose-dependent manner, an effect that was abolished after CSN resection [13]. Moreover, insulin receptors were shown to be present at the CB, and insulin, in physiological concentrations, was able to elicit a neurosecretory response in CB type I cells, measured by the increase in intracellular calcium and by the release of ATP and catecholamines [13]. When the effect of insulin, perfused intravenously, was evaluated in the output of the CB, insulin was shown to increase CSN and sympathetic nervous system electrophysiological activity in vivo in the rat, the effect on the sympathetic nervous system being abolished in CSN-sectioned animals [19,20]. In agreement with these, Barbosa et al. [67,68] observed in heathy volunteers that hyperinsulinemia, during a euglycemic clamp, increased minute ventilation independently of alterations in glucose levels. Confirming the effect of insulin via its action in the CB on glucose homeostasis, Vidal et al. [68] recently showed that insulin promotes hepatic glycogenolysis by acting on the CB. Taking all these data together, we postulate that, in metabolic diseases, hyperinsulinemia, rather than hyperglycemia, is one of the major stimuli contributing to CB dysfunction [13,14] (Figure 2). Alongside these, increased CB chemosensitivity, measured by the Dejours test, observed in prediabetes patients is directly correlated with plasma insulin levels and with insulin resistance but not with fasting glycemia [21].

## 3. Hyperleptinemia: A Major Factor Contributing to CB Dysfunction in Metabolic Diseases

Leptin is an adipocyte-derived hormone that acts on the hypothalamus to regulate food intake and energy consumption [69,70,71] and promotes lipolysis by the activation of sympathetic inputs to adipose tissue [72]. However, the scientific excitement about leptin discovery faded when it was demonstrated that plasma leptin levels were increased in obesity and metabolic diseases, defining a state of leptin resistance [69,70,71]. Indeed, both human and animal studies have demonstrated resistance to the anorexic and weight-lowering effects of leptin, while some of its actions on the sympathetic nervous system, namely renal and adrenal activation, are preserved, demonstrating a state of selective leptin resistance [73,74,75]. Apart from its role in metabolism, leptin is known to play an important role in immunity and inflammation [76], being also involved in the central control of breathing, as the administration of leptin reverses hypoxia and hypercapnia in animal models with a mutation in the leptin gene [77,78]. Moreover, leptin levels are increased in OSA patients [36,79] and correlates with OSA severity [79], suggesting that part of the CB overactivity and altered hypoxic ventilatory responses in OSA and obesity might be associated with a direct action of leptin in the CB.

Groeben et al. [80], in 2004, were the first to suggest that leptin could also act on peripheral chemoreceptors. The authors showed that hyperoxia decreased the respiratory rate in wild-type mice but not in leptin receptor deficient (ob/ob) mice, an effect that was restored by leptin replacement in ob/ob mice [80]. Later, in 2011, Porzionato et al. [81] described the presence of leptin and leptin receptor isoforms in type I, but not type II, cells of both rat and human CBs. In the CBs of humans, approximately 40% of type I cells were immunoreactive to leptin, 57% of the type I cells being immunoreactive for all leptin receptors isoforms, with approximately 30% being reactive for Ob-Rb isoforms [81]. The authors postulated that these findings were associated with a physiological role of circulating or locally produced leptin in the regulation of CB function by means of a direct action on type I cells [81]. The work of Porzionato et al. [81] was then corroborated by Messenger et al. [82] as they described that rat CB cells express the Ob-Rb and that these receptors overlap in distribution with cells expressing tyrosine hydroxylase, indicating the presence of this leptin receptor in CB type I cells. More recently, the presence of Ob-Rb isoforms was also described in mice type I and type II cells [83]. Altogether, these findings support the idea that the effects of leptin on ventilation are, at least in part, mediated by CB chemoreceptors.

In Wistar rats, intravenous [22,84] and intracarotid [22] administration of leptin increased basal ventilation and ventilation in response to ischemic hypoxia, assessed by the occlusion of the common carotid artery, in a dose-dependent manner. In the same line of evidence, subcutaneous leptin administration in C57BL/6J mice increased minute ventilation and hypoxic ventilatory response, effects abolished by the CSN resection [83]. These results, together with the findings that CSN resection decreased in approximately 30% of subjects the spontaneous ventilation induced by acute leptin intracarotid administration (90 and 270 ng/mL) (Figure 3), confirmed that the CB contributes to the effects of leptin on basal ventilation and in response to acute hypoxia. However, as described for the effects of leptin in satiety [85], the acute and chronic actions of leptin on basal ventilation seem to be opposite. Chronic leptin administration for 7 days (60 μg/mL) did not modify resting respiratory parameters but increased hypoxic ventilatory response in rats, effects abolished by CB denervation [86]. Additionally, animals submitted to a high-fat (HF) diet (60% energy from fat) for 3 weeks, although exhibiting increased basal ventilation [22], showed a 40–50% decrease in the excitatory effect of leptin in spontaneous ventilation [22] (Figure 3), effects that were not modified with CSN resection (Figure 3). In agreement with the effects of leptin as a sympathetic activator, leptin increased electrophysiological sympathetic nervous system activity, measured at the cervical sympathetic chain, in control animals, but in HF animals, the rise in sympathetic activity induced by the HF diet was not modified after leptin administration or CSN resection [86]. Alongside these blunted effects on ventilation and on the sympathetic activity produced by a HF diet and the absence of effects of CSN resection in these animals, leptin was unable to modify the increase in basal CSN electrophysiological activity induced by 3 weeks of HF diet in rats [22], while in control rats and mice, leptin augmented basal CSN chemoreceptor activity [22,64,83]. Additionally, 3 weeks of HF diet increased the expression of leptin receptor Ob-R (short and long form) by 35% [22], suggesting a feed-forward mechanism to increase CB activity or the saturation of Ob-R leptin receptors during hyperleptinemic states, in parallel to what is described in the central nervous system [87]. Taken together, these results suggest that leptin is probably involved in the activation of the CB in initial states of dysmetabolism that run with hyperleptinemia, such as overweight and prediabetes, but that a resistance to leptin signaling and the blunting of leptin responses might develop with chronic hyperleptinemia. This initial involvement of leptin in CB overactivation was also supported by data showing that CB activity is increased in both lean and obese animal models of insulin resistance but that obese animals possess more pronounced increases in spontaneous ventilation, ischemic hypoxia-induced hyperventilation, CB weight, and tyrosine hydroxylase expression at the CB compared to lean animals [13]. In addition, in concordance with the blunting produced by chronic hyperleptinemia, animals submitted to longer periods of exposure to hypercaloric diets—8 weeks of HF diet [88], 16 weeks of HF diet [89], and 25 weeks of HF-high sucrose diet [16]—exhibited decreased basal ventilation and decreased responses to hypoxia, as well as the genetic model, the obese Zucker rat, a model that lacks the gene coding for the Ob-R leptin receptor [88]. The role of CB in mediating the effects of leptin on spontaneous ventilation and hypoxic ventilatory responses in obesity was also confirmed by Caballero-Eraso et al. [83]. In Ob-R leptin receptor deficient obese *db*/*db* mice, the application of an adenovirus harboring the Ob-R gene bilaterally in the CB area promoted an increase in minute ventilation during wakefulness and sleep and augmented the hypoxic ventilatory response without affecting food intake, rectal temperature, body weight, and circulating leptin levels [83]. These results lead the authors to suggest that leptin signaling in the CB may protect against sleep disordered breathing in obesity.

Leptin is also known to be one of the key factors associated with the development of obesity-induced hypertension, acting both centrally and peripherally [3,4]. Recently, the CB was described to contribute to leptin-induced hypertension, since CSN denervation abolished the increase in mean arterial pressure induced by 3 days of subcutaneous administration of leptin in C57BL/6J mice [90]. The effect of leptin on the CB, at least in modulating blood pressure, seems to be mediated by the transient receptor potential melastatin 7 channel (Trpm 7), as its blocker, FTY720, abolished the increase in blood pressure [89], and the SKF96365, 2-APB Trp channel blockers abolished the leptin-mediated increase in CSN activity [65]. In agreement with these findings, the Trpm7 was identified in CB [64], co-localized with the Ob-R leptin receptor in CB type I cells [90], and leptin administration increased Trpm7 current in isolated CB type I cells [90]. Confirming the involvement of the CB in the development of obesity-induced hypertension, Shin et al. [89] showed that the overexpression of Ob-R leptin receptor in the CB of Ob-R-deficient obese *db*/*db* mice induced hypertension. In conclusion, leptin is able to increase CB activity, being involved in obesity and its cardiovascular consequences. The levels of circulating and/or locally produced leptin might be determinant for the initiation and maintenance of dysmetabolic states associated with obesity.

Knowing that, in obesity, and in parallel with the increased secretion of leptin from adipose tissue, there is a disturbed adipose tissue secretory pattern characterized by an increased release of pro-inflammatory factors and decreased production of anti-inflammatory adipokines [42] and associated with the development of obesity-associated comorbidities, we hypothesize that inflammation can also contribute to the CB dysfunction that is involved in the genesis of metabolic diseases.

## 4. Inflammation: A Role in CB Dysfunction?

In the last few years, a growing body of evidence established that the CB can sense both pro- and anti-inflammatory mediators. The expression of receptors for TNF-α and for the interleukins IL-1, IL-6, and IL-10 has been shown in human CBs, and as well in the mouse CB, except for the IL-1 and IL-10 receptors [91,92]. In addition, the rat CB expressed IL-1 receptor in type I cells [93,94,95], although the expression of this receptor was not restricted to these cells, since this receptor was observed in other cellular types including blood vessels, type II cells, or connective tissue cells [93]. IL-1β mRNA and immunoreactivity were also described in rat CBs and co-localized with CB type I cells [94,95,96]. Additionally, the presence of TNF-α, IL-1β, and IL-6 receptors was determined in rat CBs by Western blot (Figure 4a). The consumption of a HF diet for 3 weeks augmented the expression of the receptor IL-1R in the CB, an effect that was smaller in HF-high sucrose (HFHSu) animals, probably because these animals, when submitted to an HFHSu diet, consume more sugar that fat in their diet (Figure 4a). Additionally, preliminary results showed that the expression of TNF-R1 increased in the CB of both rat models of metabolic dysfunction, being higher in HFHSu animals, while the expression of the receptor IL-6Rα did not change with hypercaloric diet consumption (Figure 4a). These results suggest that alterations in IL-1β and TNF-α signaling might contribute to the CB overactivation described in metabolic diseases [13,22].

Besides the presence of TNF-α, IL-1β, IL-6, and their receptors in the CB, it has been described that these pro-inflammatory cytokines can modulate CB type I cells’ excitability, CB neurotransmitter release, and CSN chemosensory discharge [97,98].

### 4.1. Effect of Inflammatory Cytokines on the Carotid Body

Evidence suggests that IL-1β can activate the CB [99,100]. In isolated rat CB type I cells, the application of IL-1β decreased the outward potassium current and triggered a transient increase in [Ca^2+^]_i_, an effect that was abolished by the application of an IL-1β receptor antagonist and without any change in the release of catecholamines [99]. In agreement with these results, we have observed that 40 ng/mL of IL-1β did not modify the release of catecholamines from CBs in control and HF rats (Figure 4b). However, extracellular recordings of CSN chemosensory activity showed an increase in the CSN frequency of discharge in response to topical application of IL-1β, an effect that was partially inhibited by suramin, a P2X receptor antagonist, but not by a D_1_ and D_2_ receptor antagonist, suggesting that ATP and not dopamine mediates the CSN response to IL-1β [98]. In addition, the expression of IL-1 receptor and tyrosine hydroxylase, a rate-limiting enzyme for catecholamine synthesis in CB type I cells, increased after intraperitoneal IL-1β administration [100]. Altogether, these results clearly demonstrate that IL-1β promotes CB activation and that the increase in its levels might contribute to CB dysfunction.

The presence of IL-6 and its receptor was also found in rat CB, IL-6R being localized in type I cells [94,95,96,101]. As for IL-1 receptor, IL-6R was found to be expressed in other cellular types apart from type I cells, probably type II cells, blood vessels, and/or connective tissues [101]. In rat CB type I cells, the application of exogenous IL-6 increased [Ca^2+^]_i_ and promoted the rise in the release of catecholamines [102]. Besides the modulation of CB type I cells excitability [102], Wang et al. [101] proposed that IL-6 might modulate the survival, proliferation, and differentiation of type 1 cells in the CB. When tested on the output of CB on ventilation, the administration of IL-6 (0.5 and 5 ng/mL) in the rat femoral vein increased minute ventilation, an effect that was abolished by CSN resection, meaning that the effect of IL-6 on ventilation is mediated by the CB (Figure 4c).

Another important pro-inflammatory cytokine is TNF-α. In the rat and in the cat, the co-localization of TNF-a and its type 1 receptor (TNF-RI) in CB type I cells has been reported [94,95,103]. However, the TNF-a type 2 receptor (TNF-RII) does not co-localize with CB type I cells but has been identified in endothelial cells [104,105] and near the cell limit of nodose–petrosal–jugular ganglion complex neurons [104]. In the cat, Fernandez et al. [103] demonstrated, through in vitro recordings of the CSN activity, that TNF-α was unable to change the basal CSN chemosensory activity, although it was able to reduce the hypoxia-induced enhanced frequency of chemosensory discharge in a dose-dependent manner [103]. This inhibitory effect of TNF-α observed in the cat contrasts with the findings reported by Lam et al. [94,95] in rats, as the authors described in dissociated CB type I cells that TNF-α application induces a rise in [Ca^2+^]_i_ in response to acute hypoxia, this increase being larger in cells from the CBs of rats exposed to chronic hypoxia [95] or chronic intermittent hypoxia [95]. In addition, these effects were also described after IL-1β and IL-6 application [94,95]. This increase in type 1 cell excitability, measured as [Ca^2+^]_i_ observed by Lam et al. [94,95], is in accordance with our findings that TNF-α, when administrated in the femoral vein in a dose of 5 ng/mL, induced an increase in minute ventilation in rats (Figure 4d). This effect was not observed with lower doses of TNF-α (0.5 ng/mL) and was abolished by CSN resection, demonstrating that the effect of TNF-α on ventilation is CB-mediated (Figure 4d). Taken together, this evidence suggests that all these pro-inflammatory cytokines are able to modulate CB function via the activation of different signaling pathways and that probably different chronic inflammatory conditions will induce different alterations in TNF-α, IL-1, and IL-6 levels that will differently contribute to CB activation, namely in metabolic diseases.

### 4.2. Role of Carotid Body in Acute Systemic Inflammation

The role of CB as an immunity receptor was also suggested in animal models of sepsis syndrome induced by the administration of lipopolysaccharide (LPS) [103,104]. Topical administration of LPS on the CB as well as intravenous administration promoted histological alterations in the cat CB, by recruiting polymorphonuclear cells into the vascular bed and the stroma of the CB [103], resembling the histological features of carotid glomitis in humans [105]. In agreement with its pro-inflammatory action in the CB, intravenous administration of LPS in cats augmented the basal respiratory frequency, decreased the ventilatory chemoreflex responses to acute hypoxia, and increased basal CSN chemosensory activity, without modifying the CSN response to acute hypoxia [103]. The effect of LPS on CSN frequency of discharge and on respiratory activity was also described in mice by the administration of zymosan, which induces inflammation [106]. Finally, the tachypnoea induced by LPS was prevented by the bilateral carotid neurotomy [103], meaning that the CB is responsible for LPS’ effects on ventilation. Searching for the mechanism behind LPS action on the CB, Fernandez et al. [104] reported the presence of toll-like receptor 4 in the CB and in the nodose–petrosal–jugular ganglion complex neurons that are activated and evoke the MyD38-dependent mechanism in CB chemoreceptor neural pathway after the intraperitoneal administration of LPS. Consequently, after LPS administration, IkB degradation with NF-kB p65 nuclear translocation in CB type I cells and neurons of the nodose–petrosal–jugular ganglion complex was observed, together with an increase in pERK fraction in the CB and an increase in pERK and p-p38 MAPK fractions in the nodose–petrosal–jugular ganglion complex [104]. The activation of this mechanism after LPS administration promotes an increase in the expression of TNF-α and its receptor, TNF-RII, both at the CB and in the nodose–petrosal–jugular ganglion complex neurons [104].

Systemic inflammation also induced the production of IL-1β in the CB, since the application of zymosan in rat type I cells activated toll-like receptor 2 and the NLRP3 inflammasome, which could be the source of IL-1β in CB type I cells [106].

More recently, it was described that electrical CSN stimulation in conscious rats attenuated the innate immune response to LPS by decreasing plasma inflammatory cytokine levels such as TNF, IL-1β, and IL-6 while enhancing the levels of the anti-inflammatory cytokine IL-10, an effect that was abolished by CB denervation [107]. In addition, the anti-inflammatory effect of CSN electrical stimulation was also abolished by the administration of propranolol, a β-adrenergic antagonist, and methylatropine, a muscarinic antagonist, demonstrating that the mechanism by which CBs mediate inflammation involves both the sympathetic and the parasympathetic systems [107]. Therefore, these results suggest that CSN stimulation might be an innovative therapeutic strategy to treat inflammatory diseases mediated by cytokines, such as auto-immune diseases and sepsis, as well as the inflammation observed in metabolic dysfunction. However, we have to take into account that possibly there will be differences between the mediators, the mechanisms, and the neural circuits in response to acute situations like sepsis, chronic inflammation, chronic hypoxia, or obesity.

### 4.3. Role of Carotid Body in Chronic Inflammation

#### 4.3.1. Chronic Sustained Hypoxia-Induced Inflammation

In chronic sustained hypoxia (CSH), the low PO_2_ exposure lasts for hours to days to months or years. CSH induces gene expression, leading to profound morphological as well as biochemical changes in the CB. It has been demonstrated that CSH induces an increase in the sensitivity of CB chemoreceptors to acute hypoxia [108] and this mechanism plays a significant role in the time-dependent increase in ventilation in breathing on ascent to high altitude, which is termed ventilatory acclimatization to hypoxia (VAH) [109,110,111,112,113]. Chronic sustained hypoxia is also involved in pathological conditions; for example, in humans, CSH is associated with chronic obstructive pulmonary disease (COPD), asthma, or pulmonary fibrosis originating from pulmonary hypertension, which are clinical situations associated with inflammation, and in infants, it is associated with sudden infant death syndrome (SIDS).

Early theories and experiments on VAH have focused on changes in the pH of cerebrospinal fluid as a stimulus for central chemoreceptors. However, time-dependent changes in this parameter do not explain VAH [114] and lead to the idea of neural plasticity, i.e., central nervous system processing of afferent information is enhanced by CSH. Nowadays, it is known that plasticity occurs both in peripheral and central chemoreceptors (for a review, see [113]).

CSH has been shown to increase oxygen sensitivity in the CB, as it increases the CSN discharges in response to hypoxic tests in several species such as goats [115], cats [108,116], and rats [117,118,119]. This increase is a major factor for the increased hypoxic ventilatory response (HVR) observed with CSH, as shown by Bisgard et al. [120,121] in goats, in a preparation in which CBs are isolated from systemic circulation, and in which 6 h of isolated CB hypoxia increased ventilation above the acute HVR [122], demonstrating that the response is specific to the CB and not due to central chemoreceptors. This plasticity induced by decreased PO_2_ in the CB during the first hours explains VAH.

Humans who are native to regions with high altitudes or have lived at high altitudes for many years may have subnormal or complete loss of the ventilatory response to acute hypoxia, commonly referred to as a “blunted” response to hypoxia [123]. This effect has been attributed to a loss of responsiveness of the CB to hypoxia. Additionally, some authors have reported decreased CB hypoxic sensitivity in response to a prolonged exposure to hypoxia (3–4 weeks at an altitude of 5500 m or breathing 10% O_2_) that contributes to the decreased HVR in chronic hypoxia [124,125,126]. Associated with CSH, and therefore with the ascension to high altitudes, there is also an acclimatization in glucose homeostasis, as short-term exposures to CSH increase fasting plasma glucose [127], followed by normalization after 1 week [128], with a subsequent decline even to pre-exposure levels during the second week of CSH [129]. These adaptations in glucose homeostasis mechanisms seem to be reflected in the inverse association found between the prevalence of diabetes and obesity with altitude in the United States adult population [130]. The initial, transient hyperglycemia could be partially attributed to the increase in sympathetic nervous system activity [130] and stress hormones, mainly catecholamines that have been documented to increase with altitude [131,132] and that are in agreement with a possible overactivation of the CB.

One of the factors that might explain the increased HVR and increased fasting plasma glucose within the first few days of CSH is CB’s response the inflammation promoted by hypoxia.

Exposure for 3, 7, and 28 days to CSH (10% O_2_) increased the expression of IL-1β, Il-6, and TNF-α and the corresponding receptors in the rat CB [94]. Additionally, exposure to 380 Torr increased the mRNA expression of IL-1β, IL-6, and TNF-α in rat CBs after 1 day, levels that remain elevated after 7 days of chronic hypoxia [133]. However, Liu et al. [133] found that, after 28 days of CSH, only the levels of IL-6 remain elevated, suggesting the development of mechanisms that can contribute to the ventilatory acclimatization to hypoxia as well as to the glucose homeostasis adaptation. Moreover, the up-regulation of IL-6 was mainly observed in CB type II cells and with less extension in CB type I cells [133], suggesting that type II cells also contribute to IL-6 production in CBs. In contrast, Feng et al. [134] did not observe significant differences in CB IL-6 levels between normoxia and CSH in rabbits, which suggests that probably different inflammatory mediators could be involved in different animal species and may be involved in the response to different chronic hypoxia protocols or times of exposure.

Ibuprofen and dexamethasone, two anti-inflammatory drugs, administered for 8–10 days of CSH decreased the rise in CB IL-1β, IL-6, and TNF-α expression induced by CSH and abolished the increase in CSN frequency of discharge induced by hypoxia in these animals [132], suggesting that the effect of CSH on chemoreceptor excitability is mediated by a local immune response in the CB. Additionally, ibuprofen treatment in rats blocked the increase in HVR described in rats exposed to CSH for 7 days at 70 Torr [135]. In agreement with the studies in animals, ibuprofen treatment for 48 h at a high altitude (3800 m, PO_2_ = 90 Torr) in healthy men and women decreased HVR compared to a placebo, without any effect on basal ventilation and arterial O_2_ saturation breathing at high altitudes [135]. These results obtained in both animals and humans show that ibuprofen impairs ventilatory acclimatization to high altitudes [135,136]. However, in rats submitted to 11–12 days of chronic hypoxia (70 Torr), treatment with ibuprofen in the last 2 days was unable to reverse the ventilatory acclimatization already established by sustained hypoxia [137]. Taken together, these results suggest that, in an early phase of CSH, an adaptive inflammatory response could be important in the modulation of CB function that promotes the increase in basal ventilation as well as the altered hypoxic ventilatory responses and hyperglycemia.

#### 4.3.2. Chronic Intermittent Hypoxia-Induced Inflammation

Chronic intermittent hypoxia (CIH) is characterized by transient episodes of hypoxia of small durations. Episodic or intermittent hypoxia is associated with many pathophysiological situations including sleep apnea and apnea of prematurity. CIH leads to systemic hypertension, myocardial and brain infarctions, cognitive dysfunction, sudden death in the elderly [138,139,140], and metabolic diseases [141].

Inflammation in CIH has been proposed as one of the possible mechanisms underlying CIH-induced comorbidities such as hypertension [142] and insulin resistance [141]. Pro-inflammatory molecules are known to stimulate the sympathetic nervous system [5] by acting on brainstem and hypothalamic nuclei, such as the nucleus of the solitary tract, which is the primary site for the afferent inputs from the CB. Therefore, it is plausible to postulate that inflammation is a major determinant in the development of CIH-induced comorbidities via activation of the CB chemoreflex pathway.

The expression of IL-1β, IL-6, and TNF-α, and their respective receptors shown to be present in the CB [93,94] were increased in the CB of rats exposed to 7 days of CIH, a typical hallmark of obstructive sleep apnea [97]. In agreement, Del Rio et al. [96] described an increase in TNF-α immunoreactivity in the rat CB after 14 and 21 days of CIH, although IL-1β immunoreactivity only increased after 21 days of CIH and IL-6 immunoreactivity in rat CB was not modified after 7, 14, or 21 days of CIH [96]. The enhanced levels of TNF-α and IL-1β in rat CBs exposed to 21 days of CIH were associated with oxidative stress, since the increase in both cytokines was prevented by ascorbic acid treatment during the hypoxia protocol [143]. Contrastingly, Feng et al. [134] reported that, in the rabbit CB, IL-6 levels increased but then decreased with the increase in intermittent hypoxia frequency, which could indicate species differences and/or different intermittent hypoxia frequencies applied. Interestingly, CIH (5% O_2_, 12 times/h per 8 h) for 7, 14, or 28 days in rats did not modify systemic plasma TNF-α and IL-1β levels, only inducing a transient increase in IL-6 levels [96].

Apart from increasing the expression of IL-1β, IL-6, and TNF-α in rat CBs exposed to CIH for 7 days, it also produced an increase in the mRNA levels of NADPH oxidase subunits, an effect attenuated by anti-inflammatory drug treatment [94]. Moreover, CBs of rats exposed to 7 days of CIH also showed an increase in chemokine levels, such as monocyte chemoatractant protein-1 (MCP-1), chemokine receptor 2 (CCR2), macrophage inflammatory protein (MIP-1α and MIP-1β), and intercellular adhesion molecule (ICAM-1), supporting the notion that CIH induced local inflammation in the CB [95].

CIH exposure for 3 and 7 days in rats produced an increase in immunoreactivity to ED1, a macrophage marker, in the CB, an effect that was prevented by dexamethasone or ibuprofen treatment [95]. In contrast with these findings, rats exposed to 21 days of CIH did not exhibit any ED1 positive cells in the CB [96]. The results obtained by Del Rio et al. [96] suggest that the elevated levels of TNF-α and IL-1β observed in the CB after 21 days of CIH were not explained by the infiltration of immune cells in the CB, or by the increase in TNF-α and IL-1β systemic levels, but by an increase in local production of cytokines in the CB. As previously described in rats exposed to 21 days of CIH, ibuprofen treatment prevented the CIH-induced increase in TNF-a and IL-1β in the CB [143,144]. This anti-inflammatory drug also prevented the increase in c-fos-positive neurons in the nucleus solitary tract, the increase in hypoxic ventilatory responses, and the development of hypertension [143], suggesting that inflammatory cytokines acting within the CB are determinant in promoting CB dysfunction and the alterations in CB-brain neuro-circuits responsible for CIH-pathological conditions. However, ibuprofen failed to avoid the CB chemosensory sensitization in response to acute hypoxia [143,144], suggesting that the mechanisms underlying the CB chemosensory sensitization promoted by CIH are not associated with an increase in CB TNF-α and IL-1β levels and a distinction between the mechanisms basal vs. hypoxic alterations in CIH.

#### 4.3.3. Obesity-Induced Inflammation

Obesity is tightly connected with OSA and therefore with CIH. There is substantial evidence that hypoxia develops within the adipose tissue as the tissue mass expands, with the reduction in interstitial PO_2_ being considered to underlie the inflammatory response, as the secretion of a number of inflammation-related cytokines (IL-6, TNFα) and adipokines are upregulated by hypoxia [141,145]. In addition, it has been shown that hypoxia stimulates glucose utilization in adipocytes, with correspondent lactate production [145,146], and this produces insulin resistance in fat cells [147]. Local hypoxia within adipose tissue also stimulates the secretion of leptin, even with a normoxic systemic PO_2_ [148]. Since many of the responses of adipocytes to hypoxia are initiated at oxygen levels above the normal physiological range for adipose tissue, it can be postulated that the CBs can be important integrators of information relating blood oxygen levels and adipose tissue homeostasis and, therefore, leptin and pro-inflammatory cytokines may represent key neurohumoral mediators in the balance of CB-mediated systemic responses of adipocytes to hypoxia.

Among the adipokines secreted by the adipose tissue are growth factor-β (TGF-β), angiopoietin, and insulin-like growth factor-1 (IGF-1) which are growth and angiogenic factors; TNF-α, IL-6, and IL-1β, classified as cytokines; and complement-like factors, such as acylation stimulating protein (ASP) [149]. The normal secretion of such substances in adipose tissue is differently influenced by diverse factors such as the increase in body fat, which changes the adipocytes’ size, and hypoxia, which alters the secretion of pro-inflammatory substances into the blood and other tissues, leading to the development of systemic inflammation [150]. However, these substances are not equally secreted in all white adipose tissue depots, the consensus being that increased visceral white adipose tissue is associated with a higher risk of developing metabolic diseases such as insulin resistance and type 2 diabetes as well as cardiovascular diseases [151]. In fact, the secretion of pro-inflammatory cytokines, such as IL-6, IL-8, MCP-1, and PAI-1, among others, is higher in visceral adipose tissue, while leptin is higher in subcutaneous adipose tissue [152].

One of the first changes that takes place in adipose tissue during obesity is the increase in the number of macrophages within the tissue, which are sources of TNF-α and other pro-inflammatory cytokines [146]. Weisberg et al. [153] estimated that the percentage of macrophages in adipose tissue changes from 10% in lean mice and humans to 40% in obese humans and 50% in extreme obesity and leptin-deficient mice. Such results allowed us to conclude that the size of adipocytes could help to predict the macrophage percentage in humans, once adipocyte volume is associated with dyslipidemia, systemic insulin resistance, and with the risk of developing type 2 diabetes [154,155]. Therefore, we can hypothesize that the increased levels of pro-inflammatory cytokines released from the visceral adipose tissue [156] or by the adipose tissue surrounding the CB can act on the CB to activate it and in the long term might contribute to CB overactivation in subjects with obesity. In fact, CBs of animals submitted to hypercaloric diets showed increased expression of TNF-α and IL-1β receptors (Figure 4a), probably representing a feed-forward vicious cycle between adipose tissue inflammation and the action of pro-inflammatory cytokines in the CB that promotes adipose tissue dysfunction and metabolic diseases.

## 5. Conclusions

In conclusion, we provide substantial evidence showing that insulin, leptin, and pro-inflammatory cytokines are able to activate the CB and modulate its function and that probably hyperinsulinemia, hyperleptinemia, and high pro-inflammatory cytokine levels are determinant factors contributing to the CB overactivation that contributes to the genesis of metabolic diseases. However, there is still a huge lack of knowledge regarding the mechanisms and signal pathways by which these mediators activate and/or modify CB action and CSN activity, particularly in metabolic disease conditions. Information on insulin, leptin, and cytokine signaling pathways within the CB might be of key importance and have significant clinical relevance as it might unravel therapeutic targets for the treatment of metabolic diseases.

## Figures and Tables

**Figure 1 ijms-21-05545-f001:**
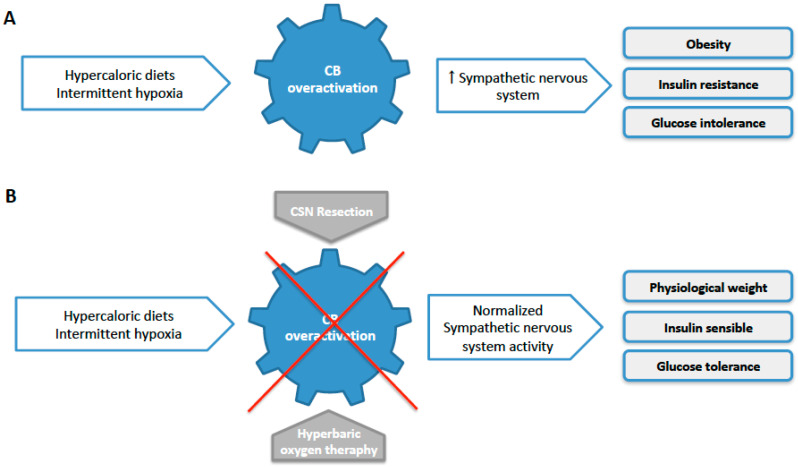
Schematic representation of the carotid body (CB) involvement in the development of obesity, insulin resistance, and glucose intolerance through an increase in sympathetic nervous system activity. (**A**) Hypercaloric diets and intermittent hypoxia promote an increase in CB activity that contributes to the augmentation of sympathetic nervous system activity, leading to metabolic dysfunction. (**B**) Modulation of CB activity through the carotid sinus nerve (CSN) resection or via hyperbaric oxygen therapy, normalized sympathetic nervous system activity, improving dysmetabolism.

**Figure 2 ijms-21-05545-f002:**
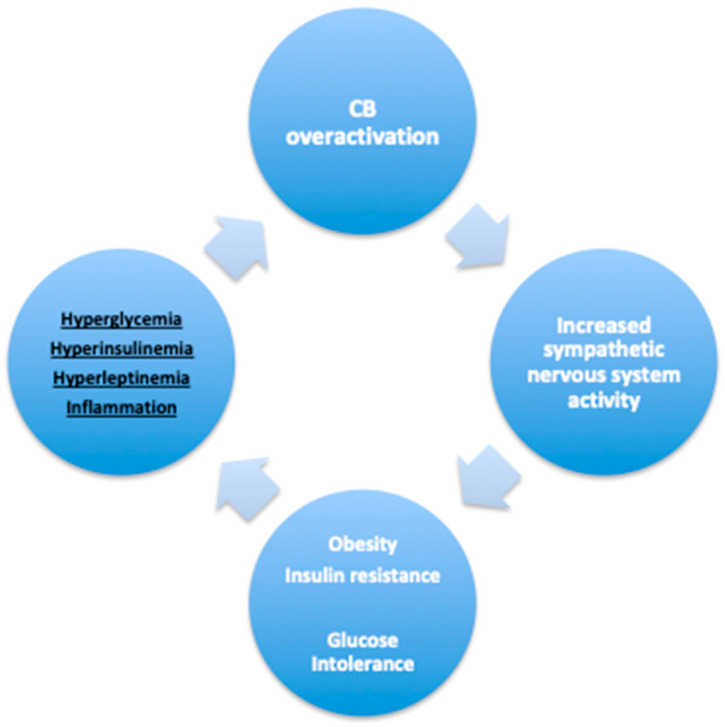
Mediators contributing to carotid body (CB) overactivation in metabolic diseases. Schematic representation of the several factors, such as hyperglycemia, hyperinsulinemia, hyperleptinemia, and inflammation, could induce an increase in CB that will contribute to the development of obesity, insulin resistance, and glucose tolerance through the overactivation of the sympathetic nervous system.

**Figure 3 ijms-21-05545-f003:**
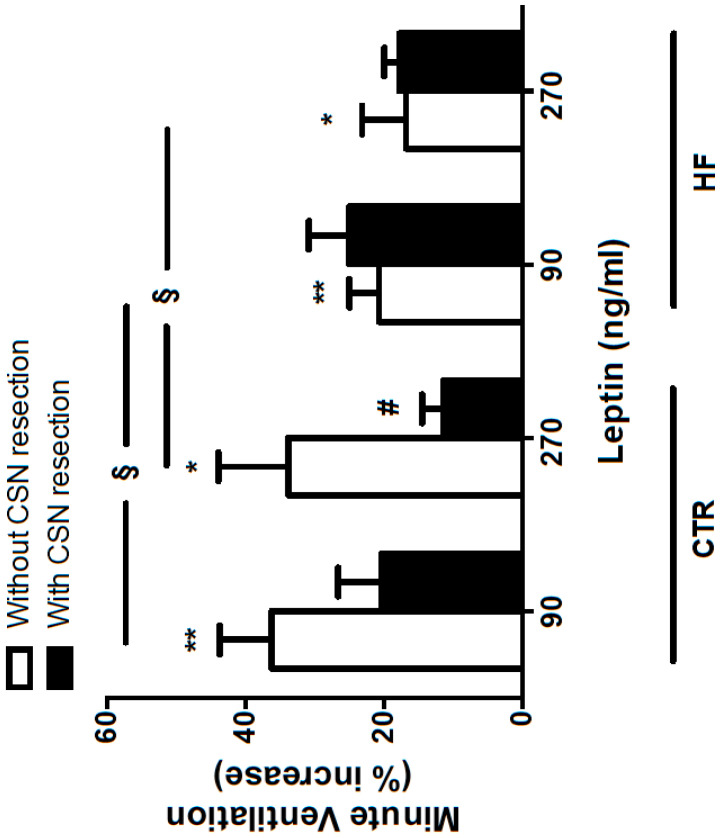
Effect of carotid sinus nerve resection (CSN) on leptin effects on ventilation in control (CTR) and high-fat (HF) animals. Leptin (90 and 270 ng/mL) was administrated intracarotidally, as a bolus in anesthetized animals with pentobarbital (60 mg/kg i.p.), as previously described by Ribeiro et al. [13,22]. CSN resection was performed acutely prior to leptin administration. HF animals were submitted to a lipid-rich diet (60% energy from fat) for 3 weeks. Data are presented as mean ± SEM of 5–8 control and 3–5 HF animals. Two-way ANOVA with Bonferroni multiple comparison tests: * *p* < 0.05, ** *p* < 0.01, leptin (ng/mL) baseline vs. diet or diet plus leptin; # *p* < 0.01 without CSN resection vs. with CSN resection; § *p* < 0.05 control vs. HF diet.

**Figure 4 ijms-21-05545-f004:**
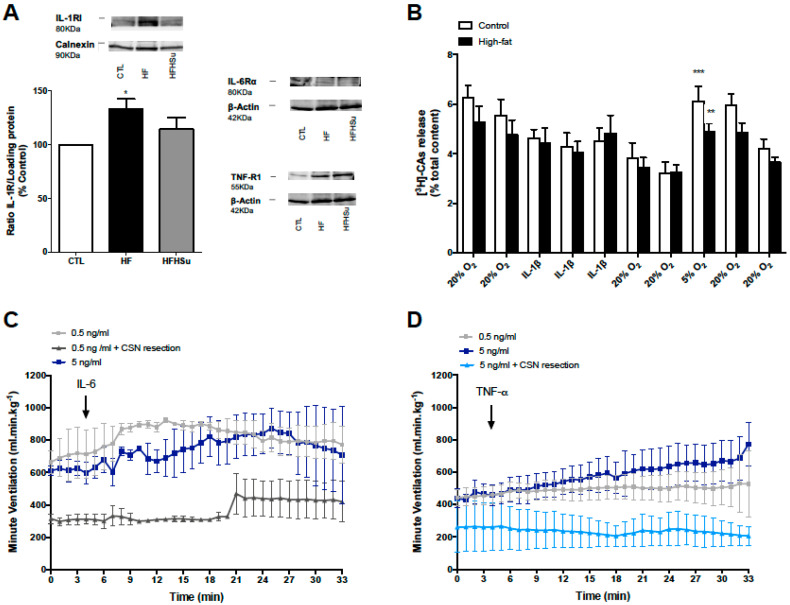
Effect of hypercaloric diets on the expression of pro-inflammatory cytokines receptors in CB and effect of pro-inflammatory cytokines on ventilation and catecholamine release from the carotid body (CB). (**A**) Effect of high-fat (HF) for 3 weeks and high-fat high-sucrose (HFHSu) diet for 25 weeks on the expression of the receptors IL-1RI, IL-6Rα, and TNF-R1 in rat CB. (**B**) Effect of interleukin-1 beta (IL-1β, 40 ng/mL) on the release of catecholamines from the CB in control and HF animals. (**C**,**D**) Effect of interleukin-6 (IL-6, 0.5 or 5 ng/mL) and tumor necrosis factor alpha (TNF-α, 0.5 or 5 ng/mL) on basal ventilation, respectively, measured in anesthetized animals with pentobarbital (60 mg/kg.i.p.). TNF-α and IL-6 were administrated in the femoral vein, as described previously by Cracchiolo et al. [20]. Carotid sinus nerve (CSN) resection was performed acutely prior to TNF-α and IL-6 administration. The catecholamine release protocol consisted of two incubations of CB in normoxic solutions (20% O_2_ plus 5% CO_2_ balanced 75% N_2_, 10 min), followed by IL-1β application for 30 min in normoxia, followed by two normoxic incubations, one hypoxic incubation (5% O_2_, 10 min), and two final normoxic incubations. The release of catecholamines from the CB was normalized for catecholamine content in each CB. Each bar represents a 10 min incubation and sample collection period. Protocol for catecholamines release from the CB was similar to that previously used [13,83]. Data are presented as mean ± SEM of 3 (**A**), 14–15 carotid bodies (**B**), and 3–6 animals (**C**,**D**). Two-way ANOVA with Bonferroni multicomparison test: * *p* < 0.05, ** *p* < 0.01 and *** *p* < 0.001 compared with 20% O_2_ prior to hypoxic (5% O_2_) stimulus.

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
