# Peer review of "Exploring the Mediators that Promote Carotid Body Dysfunction in Type 2 Diabetes and Obesity Related Syndromes"

_ijms, 2020, doi:10.3390/ijms21155545_

Round 1

Reviewer 1 Report

That is an important review which summarizes current knowledge in the field of the mediators that influences the activity of peripheral chemoreceptors in the context of metabolic diseases. Most of the subsections end with the hypotheses, which should show a direction of future studies. The main consideration regarding this review is the misuse of the definition of peripheral chemoreceptors activity and sensitivity (please follow my comments below), which must be corrected before publication. Please remember also that when in animal models (e.g. in rats) the dominant group of peripheral chemoreceptors are carotid bodies (some authors suggest even, that in rats there are no other peripheral chemosensory areas); in humans, aortic bodies have a strong influence on sympathetic activity, which may have an influence on future studies in human beings (Niewinski, Piotr, et al. "Dissociation between blood pressure and heart rate response to hypoxia after bilateral carotid body removal in men with systolic heart failure." Experimental physiology 99.3 (2014): 552-561., Tubek, Stanislaw, et al. "Effects of selective carotid body stimulation with adenosine in conscious humans." The Journal of physiology 594.21 (2016): 6225-6240.).  

Major issue:

  1. Page 2 Line 32-34

“……correlates with increased peripheral CB chemosensitivity, evaluated by the Dejours test, that measures the decrease in basal ventilation produced by 100% O2(hyperoxia)…”

and

Page 5 line 38-39

“Alongside, increased CB chemosensitivity, measured by the Dejours test…”

Dejours test shows the results of PCh (including carotid and aortic bodies) inhibition, not activation. Since this is not CB chemosensitivity, Dejours test is designed for tonic peripheral chemoreceptors activity assessment (Tubek, Stanislaw, et al. "Human carotid bodies as a therapeutic target: new insights from a clinician's perspective." Kardiologia polska 76.10 (2018): 1426-1433. or Paton, J. F., Ratcliffe, L., Hering, D., Wolf, J., Sobotka, P. A., & Narkiewicz, K. (2013). Revelations about carotid body function through its pathological role in resistant hypertension. Current hypertension reports15(4), 273-280.). Consider changing the nomenclature here.

Minor issues:

  1. Page 5 line 40

Consider analyzing the paper given below as proof of the relationship between Peripheral Chemosensitiviy (defined as increased blood pressure response to hypoxia), obesity and increased insulin levels (and insulin resistance).  

Bartłomiej Paleczny, Agnieszka Siennicka, Maciej Zacharski, Ewa Anita Jankowska, Beata Ponikowska, Piotr Ponikowski.: Increased body fat is associated with potentiation of blood pressure response to hypoxia in healthy men: relations with insulin and leptin

 Clin.Auton.Res. 2016 Vol.26 no.2; s.107-116

 https://link.springer.com/article/10.1007/s10286-015-0338-2

 DOI: 10.1007/s10286-015-0338-2

  1. Figure 1 and Figure 2

 Carotid body activation with leptin, inflammatory factors, etc leading to increased sympathetic activation closes the vicious circle of metabolic diseases. One could understand when analysing the figures, that CB deactivation may prevent/treat obesity and metabolic diseases. I would recommend changing these chain diagrams to circular diagrams. 

  1. Figure 4

Figure 3. –Effect of carotid sinus nerve resection (CSN) on leptin effects in ventilation in control (CTR) and high-fat (HF) animals

Should be:

Figure 3. –Effect of carotid sinus nerve resection (CSN) on leptin effects on ventilation in control (CTR) and high-fat (HF) animals

  1. Figure 4

Please clarify the description of Fig. 4b or add a timeline to the X-axis. The current layout is difficult to understand.

  1. Page 9 line 39-41  

In the rat and in the cat, the co-localization of TNF-aand its type I receptor, TNF-RI, in CB type I cells has been reported [95,96,104]. However, the TNF-atype 2 receptor, TNF-RII, do not co-localize with CB type I cells, but has been identified in endothelial cells…..

Should be:

In the rat and in the cat, the co-localization of TNF-a and its type 1 receptor (TNF-RI) in CB type I cells has been reported [95,96,104]. However, the TNF-a type 2 receptor (TNF-RII) do not co-localize with CB type I cells, but has been identified in endothelial cells……

  1. Page 10, line 41-43

However, we have to take into account that possibly there will be differences between the mediators, the mechanisms and the neural-circuits in response to situations of acute, like sepsis and chronic inflammation, as chronic hypoxia and obesity.

Should be:

However, we have to take into account that possibly there will be differences between the mediators, the mechanisms and the neural-circuits in response to acute situations, like sepsis and chronic inflammation, and chronic hypoxia or obesity.

  1. Page 11, line 2

Chronic sustained hypoxia is also involved in pathological conditions, for example, in humans CSH is associated with chronic obstructive pulmonary disease (COPD), asthma or pulmonary fibrosis originating pulmonary hypertension, and in infants it is associated with sudden infant death syndrome (SIDS), clinical situations that are associated with inflammation.

Should be:

Chronic sustained hypoxia is also involved in pathological conditions, for example, in humans, CSH is associated with chronic obstructive pulmonary disease (COPD), asthma or pulmonary fibrosis originating pulmonary hypertension, which are clinical situations associated with inflammation, and in infants, it is associated with sudden infant death syndrome (SIDS).

  1. Page 12, line 47

As previously described in rats exposed to chronic hypoxia, ibuprofen treatment in rats exposed to 21 days of CIH prevented the CIH-induced increased TNF-aand IL-1bin the CB

Should be:

As previously described in rats exposed to 21 days of CIH, ibuprofen treatment prevented the CIH-induced increased TNF-a and IL-1bin the CB

Author Response

That is an important review which summarizes current knowledge in the field of the mediators that influences the activity of peripheral chemoreceptors in the context of metabolic diseases. Most of the subsections end with the hypotheses, which should show a direction of future studies. The main consideration regarding this review is the misuse of the definition of peripheral chemoreceptors activity and sensitivity (please follow my comments below), which must be corrected before publication. Please remember also that when in animal models (e.g. in rats) the dominant group of peripheral chemoreceptors are carotid bodies (some authors suggest even, that in rats there are no other peripheral chemosensory areas); in humans, aortic bodies have a strong influence on sympathetic activity, which may have an influence on future studies in human beings (Niewinski, Piotr, et al. "Dissociation between blood pressure and heart rate response to hypoxia after bilateral carotid body removal in men with systolic heart failure." Experimental physiology 99.3 (2014): 552-561., Tubek, Stanislaw, et al. "Effects of selective carotid body stimulation with adenosine in conscious humans." The Journal of physiology 594.21 (2016): 6225-6240.).  

The authors acknowledge the reviewer’s comments and precious inputs on the manuscript as they feel that his/her comments contribute to increase the quality of the manuscript.

Major issue:

  1. Page 2 Line 32-34

“……correlates with increased peripheral CB chemosensitivity, evaluated by the Dejours test, that measures the decrease in basal ventilation produced by 100% O2(hyperoxia)…”

and

Page 5 line 38-39

“Alongside, increased CB chemosensitivity, measured by the Dejours test…”

Dejours test shows the results of PCh (including carotid and aortic bodies) inhibition, not activation. Since this is not CB chemosensitivity, Dejours test is designed for tonic peripheral chemoreceptors activity assessment (Tubek, Stanislaw, et al. "Human carotid bodies as a therapeutic target: new insights from a clinician's perspective." Kardiologia polska 76.10 (2018): 1426-1433. or Paton, J. F., Ratcliffe, L., Hering, D., Wolf, J., Sobotka, P. A., & Narkiewicz, K. (2013). Revelations about carotid body function through its pathological role in resistant hypertension. Current hypertension reports15(4), 273-280.). Consider changing the nomenclature here.

We have changed accordingly with reviewer comment.

Minor issues:

  1. Page 5 line 40

Consider analyzing the paper given below as proof of the relationship between Peripheral Chemosensitiviy (defined as increased blood pressure response to hypoxia), obesity and increased insulin levels (and insulin resistance).  

Bartłomiej Paleczny, Agnieszka Siennicka, Maciej Zacharski, Ewa Anita Jankowska, Beata Ponikowska, Piotr Ponikowski.: Increased body fat is associated with potentiation of blood pressure response to hypoxia in healthy men: relations with insulin and leptin

 Clin.Auton.Res. 2016 Vol.26 no.2; s.107-116

 https://link.springer.com/article/10.1007/s10286-015-0338-2

 DOI: 10.1007/s10286-015-0338-2

We appreciated the reviewer input to our manuscript and we have cited the suggested paper.

  1. Figure 1 and Figure 2

 Carotid body activation with leptin, inflammatory factors, etc leading to increased sympathetic activation closes the vicious circle of metabolic diseases. One could understand when analysing the figures, that CB deactivation may prevent/treat obesity and metabolic diseases. I would recommend changing these chain diagrams to circular diagrams. 

 We have changed Figure 1 accordingly with the suggestions from reviewer 1 and 2.

  1. Figure 4

Figure 3. –Effect of carotid sinus nerve resection (CSN) on leptin effects in ventilation in control (CTR) and high-fat (HF) animals

Should be:

Figure 3. –Effect of carotid sinus nerve resection (CSN) on leptin effects on ventilation in control (CTR) and high-fat (HF) animals

 Corrected accordingly with  reviewer suggestion.

  1. Figure 4

Please clarify the description of Fig. 4b or add a timeline to the X-axis. The current layout is difficult to understand.

Information about the protocol was included in figure legend. We hope that this could help to understand clearer the figure.

  1. Page 9 line 39-41  

In the rat and in the cat, the co-localization of TNF-aand its type I receptor, TNF-RI, in CB type I cells has been reported [95,96,104]. However, the TNF-atype 2 receptor, TNF-RII, do not co-localize with CB type I cells, but has been identified in endothelial cells…..

Should be:

In the rat and in the cat, the co-localization of TNF-a and its type 1 receptor (TNF-RI) in CB type I cells has been reported [95,96,104]. However, the TNF-a type 2 receptor (TNF-RII) do not co-localize with CB type I cells, but has been identified in endothelial cells……

Done.

  1. Page 10, line 41-43

However, we have to take into account that possibly there will be differences between the mediators, the mechanisms and the neural-circuits in response to situations of acute, like sepsis and chronic inflammation, as chronic hypoxia and obesity.

Should be:

However, we have to take into account that possibly there will be differences between the mediators, the mechanisms and the neural-circuits in response to acute situations, like sepsis and chronic inflammation, and chronic hypoxia or obesity.

Done.

  1. Page 11, line 2

Chronic sustained hypoxia is also involved in pathological conditions, for example, in humans CSH is associated with chronic obstructive pulmonary disease (COPD), asthma or pulmonary fibrosis originating pulmonary hypertension, and in infants it is associated with sudden infant death syndrome (SIDS), clinical situations that are associated with inflammation.

Should be:

Chronic sustained hypoxia is also involved in pathological conditions, for example, in humans, CSH is associated with chronic obstructive pulmonary disease (COPD), asthma or pulmonary fibrosis originating pulmonary hypertension, which are clinical situations associated with inflammation, and in infants, it is associated with sudden infant death syndrome (SIDS).

Done.

  1. Page 12, line 47

As previously described in rats exposed to chronic hypoxia, ibuprofen treatment in rats exposed to 21 days of CIH prevented the CIH-induced increased TNF-aand IL-1bin the CB

Should be:

As previously described in rats exposed to 21 days of CIH, ibuprofen treatment prevented the CIH-induced increased TNF-a and IL-1bin the CB

Done.

Reviewer 2 Report

Main comments

The review is interesting reading that rather comprehensively describes the relatively less known chemoreceptive properties of carotid bodies (CB) apart from its main oxygen/CO2/pH sensing. However, in my opinion, the message of the article is not fully reflected by the title and abstract due to the following reasons:

  1. The main message of the review as the authors define it themselves is the role of the carotid body in metabolic disorders. However, the data presented reflects mainly CB involvement in the pathogenesis of type 2 diabetes and related obesity syndrome.
  2. The second part of the review is dedicated mostly to the involvement of CB in the inflammatory response, specifically under the various types of hypoxia.
  3. Despite the known presence of the inflammatory component in metabolic disorders, the connection as presented in the review appears to be relatively weak (for instance, in Fig. 4).
  4. Authors discuss the role of CB mostly in terms of its dysfunction, however, it should be noted that, especially in case of the inflammatory response at the initial stages, CBs apparently execute an adaptive, homeostatic function.

Therefore, I would suggest modifying the title and also the abstract to make them more inclusive.

It’s not common to present original data (which I assume is the data in figures 3 and 4) in the review paper. However comprehensive the figure legend is it is still difficult to evaluate such data without a proper Methods section.

Minor comments

  1. Page 2, line 8. Energy homeostasis is a very broad term that includes not only the metabolism of carbohydrates and lipids where the CBs seem to be involved most.
  2. In the panel B of Fig. 1, the inhibition of CB activity, as authors suggest, restores the activity of the sympathetic nervous system with the corresponding recovery of the metabolic dysfunction. This must be reflected in the figure as well, for instance by changing the direction of the arrow and corresponding changes for metabolic disturbances.
  3. Page 3, lines 13-16. The statement concerning the causative role of CB in the OSA hypertension appears to be too conclusive and not directly inferred from the cited references.
  4. Page 4, line12. Correct misspelled “citokines” and also few grammatic/syntactic errors elsewhere in the text.

Author Response

The review is interesting reading that rather comprehensively describes the relatively less known chemoreceptive properties of carotid bodies (CB) apart from its main oxygen/CO2/pH sensing.

The authors acknowledge the reviewer’s comments and precious input on the critical discussion of the manuscript as they feel that reviewer’s comments contributed to increase the quality of the manuscript. We hope that our answers pleased the reviewer.

However, in my opinion, the message of the article is not fully reflected by the title and abstract due to the following reasons:

  1. The main message of the review as the authors define it themselves is the role of the carotid body in metabolic disorders. However, the data presented reflects mainly CB involvement in the pathogenesis of type 2 diabetes and related obesity syndrome.

We have changed the title of the manuscript according with the reviewer indication.

  1. The second part of the review is dedicated mostly to the involvement of CB in the inflammatory response, specifically under the various types of hypoxia.
  2. Despite the known presence of the inflammatory component in metabolic disorders, the connection as presented in the review appears to be relatively weak (for instance, in Fig. 4).

Inflammation and hypoxia are considered to be main factors contributing to adipose tissue dysfunction, one of the major factors contributing to dysmetabolism. We have now included a more detailed description of the links between inflammation and metabolic disorders. Please see section 3.3.

  1. Authors discuss the role of CB mostly in terms of its dysfunction, however, it should be noted that, especially in case of the inflammatory response at the initial stages, CBs apparently execute an adaptive, homeostatic function.

The reviewer is right and in all situations - normoleptinemia, normoinsulinemia and also in response to an inflammatory response at the initial stages - the CBs play an adaptive, homeostatic function. In fact, the authors believe that the physiological function of the organ is to be an adaptive, homeostatic regulator of metabolism. However, in conditions of chronic stimulation, as it happens with hypercaloric diets and chronic hypoxia it becomes deregulated. This concept is not new, as it also happens with the sympathetic nervous system. The present manuscript deals with the factors that contribute to CB dysfunction focusing the particular case of metabolic diseases, although at the beginning of each section the effect of leptin, insulin and inflammation on the CB in physiological conditions is described.

Therefore, I would suggest modifying the title and also the abstract to make them more inclusive.

The title and the abstract were changed accordingly.

It’s not common to present original data (which I assume is the data in figures 3 and 4) in the review paper. However comprehensive the figure legend is it is still difficult to evaluate such data without a proper Methods section.

We know that is not common to include original data in a review paper, but we had permission from the Special issue” Neurotransmitters and Neuropeptides in the Modulation of the Carotid Body” Editors. We have included a more detailed description of the methods in figure legend.

Minor comments

  1. Page 2, line 8. Energy homeostasis is a very broad term that includes not only the metabolism of carbohydrates and lipids where the CBs seem to be involved most.

We have changed the words according with the reviewer comment. Now it can be read “Besides its role as an oxygen sensor, in the last years, the CB has also been proposed to be a metabolic sensor implicated in the control of cardohydrates and lipids metabolism (…)

  1. In the panel B of Fig. 1, the inhibition of CB activity, as authors suggest, restores the activity of the sympathetic nervous system with the corresponding recovery of the metabolic dysfunction. This must be reflected in the figure as well, for instance by changing the direction of the arrow and corresponding changes for metabolic disturbances.

The reviewer is right and we changed the figure accordingly.

  1. Page 3, lines 13-16. The statement concerning the causative role of CB in the OSA hypertension appears to be too conclusive and not directly inferred from the cited references.

The sentence was changed and now it can be read: “It is consensual that CB overactivation due to chronic intermittent hypoxia (CIH), is in the genesis of OSA-mediated hypertension and insulin resistance through an increase in sympathetic nervous system activity [10,26,28,29,34].”

  1. Page 4, line12. Correct misspelled “citokines” and also few grammatic/syntactic errors elsewhere in the text.

The misspelled word “citokines” as well as the grammatical errors were corrected throughout the manuscript.

Reviewer 3 Report

This is a comprehensive review of an important problem that builds on the recent findings from the group and others. My suggestions for your consideration /improvement are as follows:

Major Comments

 Much evidence supporting the role of carotid body in metabolic diseases is based on animals experiments involving carotid sinus nerve resection. It should be noted that carotid sinus nerve resection itself also reduces baroreflex sensitivity, which is an important regulator of sympathetic activity. Authors should comment the limitation of CSN resect in studying the role of carotid body in metabolic diseases.

Although it has been reported that insulin, leptin, hypoglycemia, hyperglycemia, and inflammatory cytokines can modify carotid sinus nerve activity in vivo, there is no solid evidence to support that they influence carotid body sensory activity in the same way when studying in vitro. Authors should comment the disconnection between in vivo and in vitro studies.

Section “3.3.1. Chronic sustained hypoxia-induced inflammation” is somewhat confusing. Given the controversy regarding carotid body hypoxic sensitivity in response to chronic hypoxia, and the fact that hypoxia decreases metabolic rate in rodents, I am not convinced that carotid body plays any role in glucose homeostasis caused by chronic hypoxia. Authors may consider revising this section, or deleting it.

Minor Comments

Page 5, lines 23-24, the meaning of the sentence  “…suggesting that the effects of insulin-induced hypoglycemia were mediated by insulin” is not clear.

Authors spelled both hypoglycemia and hypoglycaemia. Please be consistent throughout the manuscript.

Author Response

This is a comprehensive review of an important problem that builds on the recent findings from the group and others. My suggestions for your consideration /improvement are as follows:

 We appreciate the reviewer comments about our work . We hope that our answers please the reviewer.

Major Comments

 Much evidence supporting the role of carotid body in metabolic diseases is based on animals experiments involving carotid sinus nerve resection. It should be noted that carotid sinus nerve resection itself also reduces baroreflex sensitivity, which is an important regulator of sympathetic activity. Authors should comment the limitation of CSN resect in studying the role of carotid body in metabolic diseases.

The reviewer is right and we are aware of the potential adverse effects of CSN resection (Conde SV. 2018. Ablation of the carotid bodies in disease: meeting its adverse effects. J Physiol). However, bilateral CSN resection is and it was, in the past, used in animal models of metabolic diseases as proof-of-principle that the CB is involved in the genesis of metabolic diseases and that the CB can be targeted to treat these diseases. In fact, several groups are trying to develop strategies to diminish CB activity and my group is an example, as we are trying to develop a bioelectronic approach for modulating CB activity (Sacramento et al. 2018, Diabetologia). Although we believe that this is an important issue that deserves to be discussed we have opted to not include it in the manuscript as it is a little bit outside the focus of the present publication and also because in the same Special Issue of the International Journal of Molecular Sciences, other authors have already dedicated to explore that theme “Carotid Body and Metabolic Syndrome: Mechanisms and Potential Therapeutic Targets. Lenise J. Kim * and Vsevolod Y. Polotsky.

Although it has been reported that insulin, leptin, hypoglycemia, hyperglycemia, and inflammatory cytokines can modify carotid sinus nerve activity in vivo, there is no solid evidence to support that they influence carotid body sensory activity in the same way when studying in vitro. Authors should comment the disconnection between in vivo and in vitro studies.

From my point of view there is no dissociation between in vitro and in vivo CSN electrophysiological studies. We have recently published the effect of insulin on CSN in vivo (Cracchiolo et al. IEEE Trans Neural Syst Rehabil Eng. 2019 Oct;27(10):2034-2043) and showed that insulin increase CSN activity. In accordance with these results, we also have unpublished data obtained in ex vivo preparations CB-CSN supporting these results.

Additionally, for the leptin effects on the CSN, Caballero-Eraso et al. (JPhysiol, 2019) showed in vivo that leptin increased baseline carotid sinus nerve activity and we showed the same in the ex vivo CSN-CB preparation (Ribeiro et al. 2018. J Physiol).

Section “3.3.1. Chronic sustained hypoxia-induced inflammation” is somewhat confusing. Given the controversy regarding carotid body hypoxic sensitivity in response to chronic hypoxia, and the fact that hypoxia decreases metabolic rate in rodents, I am not convinced that carotid body plays any role in glucose homeostasis caused by chronic hypoxia. Authors may consider revising this section, or deleting it.

We agree with the reviewer, however as we state at the end of this section of the manuscript we  trust that inflammation is involved in an early phase of CSH promoting the increase in basal ventilation, as well as, the altered hypoxic ventilatory responses and hyperglycemia. We have changed the last sentence of this section and nnow it can be read:  “Taken together, these results suggest in an early phase of CSH an adaptive inflammatory response could be important in the modulation of CB function that promotes the increase in basal ventilation, as well as, the altered hypoxic ventilatory responses and hyperglycemia.”

Minor Comments

Page 5, lines 23-24, the meaning of the sentence  “…suggesting that the effects of insulin-induced hypoglycemia were mediated by insulin” is not clear.

 Sentence was modified to “Bin-Jaliah et al. [63] described that insulin-induced hypoglycemia increased spontaneous ventilation in rats, an effect that was abolished in CSN sectioned animals, but that hypoglycaemia per se was unable to alter CSN frequency, suggesting that the effects of insulin-induced hypoglycemia were mediated by only by insulin and not by hypoglycemia.”

Authors spelled both hypoglycemia and hypoglycaemia. Please be consistent throughout the manuscript.

We have changed all to hypoglycemia.